# Green Chemistry Principles for Nano- and Micro-Sized Hydrogel Synthesis

**DOI:** 10.3390/molecules28052107

**Published:** 2023-02-23

**Authors:** Sonia Trombino, Roberta Sole, Maria Luisa Di Gioia, Debora Procopio, Federica Curcio, Roberta Cassano

**Affiliations:** Department of Pharmacy, Health and Nutritional Science, University of Calabria, 87036 Arcavacata, Italy

**Keywords:** hydrogels, green chemistry, biopolymers, drug delivery, click reaction, cross-linking

## Abstract

The growing demand for drug carriers and green-technology-based tissue engineering materials has enabled the fabrication of different types of micro- and nano-assemblies. Hydrogels are a type of material that have been extensively investigated in recent decades. Their physical and chemical properties, such as hydrophilicity, resemblance to living systems, swelling ability and modifiability, make them suitable to be exploited for many pharmaceutical and bioengineering applications. This review deals with a brief account of green-manufactured hydrogels, their characteristics, preparations, importance in the field of green biomedical technology and their future perspectives. Only hydrogels based on biopolymers, and primarily on polysaccharides, are considered. Particular attention is given to the processes of extracting such biopolymers from natural sources and the various emerging problems for their processing, such as solubility. Hydrogels are catalogued according to the main biopolymer on which they are based and, for each type, the chemical reactions and the processes that enable their assembly are identified. The economic and environmental sustainability of these processes are commented on. The possibility of large-scale processing in the production of the investigated hydrogels are framed in the context of an economy aimed at waste reduction and resource recycling.

## 1. Introduction

In recent years, hydrogels have been the object of study both because they are capable of retaining a high water content and because they are very similar to the native extracellular matrix (ECM). They are used in protein transport [1,2] and tissue engineering [3], and as substrates for cell cultures [4]. Bulk hydrogels are crosslinked in continuous volumes of millimetre dimensions or greater. Using various techniques, such as cryogel formation, porogen leaching [5,6] or electrospinning, micrometre-scale porosity can be introduced into a bulk hydrogel [7]. However, bulk hydrogels are not always usable in the applications for which they were designed because smaller sizes are needed. For this reason, to obtain hydrogels such as microscale MHPs or nanoscale NP particles (microgel or nanogel), various techniques have been formulated.

The first person to use the term “nanotechnology” was Taniguchi in 1974 [8], who gave the following definition: “nanotechnology mainly consists of the processing of separation, consolidation, and deformation of materials by one atom or one molecule”. However, the theoretical basis of nanotechnology is owed to Nobel Prize winner Richard Feynman, fifteen years earlier, in his famous lecture ‘There’s a Lot of Space at the Bottom’, delivered on 29 December 1959 at the California Institute of Technology (Caltech) [9]. Therefore, Feynman is considered the father of modern nanotechnology. Nanotechnology concerns the manipulation, reduction and fabrication of nanoscale materials with sizes between 1 and 100 nm and with peculiar characteristics, such as good strength, high stability, cost-effectiveness, biocompatibility, etc. The low volume-to-surface ratio and the small dimensions of nanoparticles (NPs) result in specific peculiarities in their chemistry and physical properties. These peculiarities have suggested their use as a powerful drug delivery system for the prevention and therapy of many diseases. A considerable amount of research has focused pm hydrogel nanoparticles (NPs) (referred to as nanogels) due to the research interest in studying these systems for their use as drug carriers.

Nanoparticle hydrogel materials show the separate characteristics possessed by hydrogels and NPs at the same time. Therefore, pharmaceutical applications can take advantage not only of the hydrophilicity, flexibility, versatility, high water absorption capacity and biocompatibility of hydrogels, but also of the benefits of NPs, such as the possibility of targeting, overcoming biological barriers, the desired biophase, e.g., to tumour sites. 

Research, in addition to the usually employed synthetic polymers, is highly centred on the production of nanohydrogels based on natural polymers. To enhance the quality of hydrogel NPs and make them targeted for specific applications through a sustainable production process, a green approach to their synthesis is required, based on the twelve principles of Green Chemistry [10,11]. The twelve principles of Green Chemistry were developed in 1991 by Paul T. Anastas, an organic chemist working in the EPA’s (Environmental Protection Agency’s, Washington, DC, USA) Pollution Prevention and Toxins Bureau, and John C. Warner [11,12,13]. These principles can be grouped into two main strands: ‘Risk Reduction’ and “Minimal Environmental Impact”, whereas the term Green Chemistry was coined for the first time in 1996, indicating a conception of chemistry aimed at steering the traditional chemical industry along sustainable paths.

The goal of Green Chemistry is to reduce “risk” in all stages of the life cycle. “Risk” is anything that can cause consequences averse to humans or the environment. The researcher, when planning the design of a chemical compound or chemical process, must make sure to minimize risk at all levels of the process. Physical dangers (e.g., flammability, explosion), global dangers (e.g., stratospheric ozone depletion) and, above all, toxicity, must be avoided. A source of risk is the nature of the chemicals used in the reaction and their products. If researchers adhere to the twelve principles in their studies, they can, if not eliminate, at least reduce the inherent hazards of chemical products and processes. Today, the principles have been summarized into the more convenient and incisive acronym PRODUCTIVELY [13]: Prevention; Renewable materials; Omit derivatization steps; Degradable chemical products; Use safe synthetic methods; Catalytic reagent; Temperature, pressure ambient; In-process monitoring; Very few auxiliary substances; E-factor, maximise feed in product; Low toxicity of chemical products; Yes, it is safe (Figure 1).

A new frontier for recycling [14], “Food Supply Chain Waste” (FSCW), is an ex-citing opportunity to develop technologies and strategies for the sustainable development of food waste. Food waste occurs after food has spoiled or expired due to poor stock management or neglect. It can also occur due to malfunctioning food production or management and technical problems, such as a lack of proper storage facilities, cold chain and packaging. 

Interest has concentrated on biofuels, whereas high-value molecules (e.g., proteins, polysaccharides, lipids, etc.) can be obtained from food waste. As an example, chitin is a polysaccharide extractable from the exoskeletons of insects [15] and molluscs. It can also be found in the cell walls of fungi [16,17]. Another example is that of cellulose extracted from biomass materials [18,19,20,21,22,23,24]. However, the hydrogel processing that follows this eco-friendly production is not always coupled with the same environmental sustainability. Indeed, both the extraction of molecules and the production of hydrogels present some critical issues. The main limitations of extraction are determined by the operating conditions: high temperatures, strong acids and bases and large quantities of water for washing. The preparation of hydrogels has a great environmental impact arising from the consumption of toxic and, consequently, environmentally unfriendly reagents and solvents. 

Unconventional strategies based on the use of green solvents, alternative heating sources (microwaves) and the use of nontoxic natural crosslinkers can limit the drawbacks in hydrogel processing. This review aims to highlight hydrogels based on the most abundant and available natural polymers. Attention has been given to preparation processes that are free of solvents and toxic chemical reagents. In contrast, materials produced by methodologies that contravene the twelve principles of Green Chemistry and are not environmentally friendly, resulting in the waste of important resources such as water, have not been included. It is easily foreseeable that in the near future, research in the field of hydrogels for biomedical applications will increasingly turn toward the exploitation of biopolymers obtained through methodologies that avoid wasting resources and respect the environment.

## 2. Hydrogels

Hydrogels are 3D structures consisting of hydrophilic polymer networks. They can adsorb water and biological fluids in high quantities (up to 1000 times their dry weight) [25].

With respect to the type of crosslinking, hydrogels can be classified into “physical” and “chemical” gels (Figure 2).

They are classified as “reversible” or “physical” if crosslinking occurs through the establishment of weak physical interactions between the main chains: hydrogen bond, ionic interactions and Van der Waals [26]. These materials are unstable and reversible and retain their integral structure with difficulty and dissolve easily when environmental factors such as temperature, pH, etc., change. 

Hydrogels are classified as “permanent” or “chemical” when they form covalent bonds between the polymer chains [27]. As a result, these types of hydrogels retain their structure even after swelling. Chemical hydrogels can be generated through crosslinking polymers using radiation, polyfunctional compounds, free-radical-generating compounds and, recently, enzymatic crosslinking [28]. They have better chemical, mechanical and thermal stability than physical hydrogels. Moreover, they can be formed from either synthetic or natural polymers [29]. The latter are biocompatible, biodegradable, bacteriostatic and even have healing properties. Synthetic hydrogels do not have these inherent bioactive properties (Figure 3).

The drugs can be loaded into hydrogels in two different ways [28]. In the first method, the drug is added to the precursor polymer solution and the gelation of the drug within the matrix is then allowed to take place. The second approach involves the hydrogel swelling in a drug solution to swelling equilibrium. Next, it is important to calculate the encapsulation capacity of the drug by the hydrogel before starting drug release.

## 3. Cellulose-Based Hydrogels

The versatility of cellulose-based hydrogels (CBHYs) has recently prompted researchers around the world to develop new materials to be exploited in healthcare. CBHYs possess biocompatibility and similarity to living tissue, which provides wide opportunities for drug release, healing of wounds and tissue repair. Cellulose is the most abundant natural material as it can be found in trees, plants, fruits, vegetables and organic waste. Therefore, it is readily available and renewable, and this is an important aspect of any research to be developed and applied. It is a polysaccharide consisting of a linear chain of β-linked D-glucose units (1/4). Cellulose is a hydrophilic material. It has inter- and intramolecular hydrogen bonds and van der Waals forces that cause many problems regarding its dissolution [18]. Research has followed several directions to obtain methods for extracting cellulose from organic waste. In the extracted material, cellulose is structured into fibrils, surrounded by a matrix of lignin and hemicellulose. Among these methods, the steam explosion, chlorine-free and ionic liquid methods are notable. Each method has its limitations concerning the high costs involved in its large-scale implementation. 

### 3.1. Extraction of Cellulose for the Production of Sustainable Hydrogels

The production of sustainable and green cellulose hydrogels requires a great deal of research work [30]. As an example, we can highlight the efforts that have been made to extract cellulose from agricultural biomass [19,31,32] (Figure 4), to engineer hydrogels based on sustainable cellulose acetate [33] and to produce monolithic cellulose/alginate hydrogels for environmental applications [34].

Lignocellulosic biomass materials (LBMs), present both in plants with woody stems and in non-woody-stemmed plants, are mainly formed by cellulose, hemicellulose and lignin. The palm oil industry is one of the main manufacturing industries in Malaysia [20,21,22,23] and is an important font of LBMs. As a residue of this processing, Oil Palm Empty Fruit Bunches (OPEFBs) is abundantly produced. Being non-woody, OPEFBs are cheap, so it is presented as a raw material with interesting and potentially useful properties for various industrial processes, representing a valid alternative to woody plants, which are expensive [24]. OPEFBs are composed of 37.3–46.5% cellulose, 25.3–33.8% hemicellulose and 27.6–32.5% lignin [35]. This high cellulose amount makes OPEFBs an excellent source to be exploited to produce a polymer’s hydrogels in a sustainable way.

The cellulose fibres are entrapped in amorphous hemicellulose and lignin. Their extraction can occur through a procedure adapted from a study conducted by Nazir et al. [36], in which the depolymerization of hemicellulose and delignification are performed through the use of solvents such as formic acid and hydrogen peroxide at low concentrations. 

Thermoresponsive cellulose hydrogels were prepared through a cold method by the incorporation of cellulose extracted from OPEFBs and Pluronic F127 polymer (PF127). Their performances were evaluated. The hydrogel with PF127 content of 20 *w*/*v*% and OPEFB cellulose content of 3 *w*/*v*% showed better performance in terms of degradation percentage ratio, swelling and in vitro release of incorporated silver sulfadiazine (SSD) [24]. The developed procedure to obtain thermos-responsive cellulose hydrogel from OPEFBs has supplied a model to exploit abundantly available agricultural biomass to produce sustainable drug-delivery systems. Podzil et al., in their review [22], described how cellulose extracted from OPEFBs can be exploited for various applications. Moreover, they have illustrated how nanocellulose produced by raw materials is introduced thanks to the hydrogen bonds that are created among different polymeric matrices and functional groups present in the same cellulose chain.

### 3.2. Nanocellulose

A single cell of natural plant fibres is formed by cellulose microfibrils. Microfibrils are formed by elementary fibrils (nanofibrils). The surface free energy reduction is at the basis of the process that causes the fibrils to group [35]. The extraction procedure is somewhat equivalent to a disaggregation, resulting in nanometre-sized cellulose. The term “nanocellulose” refers to three types of nanosized cellulosic extracts: cellulose nanofibers (CNFs), cellulose nanocrystals (CNCs) and bacterial nanocellulose (BNC), each having different dimensions and properties [37]. CNFs and CNCs can be extracted from OPEFB.

In 1951, Ranby Bengt published the first experimental work about the production of colloidal cellulose suspensions by sulfuric acid degradation of cellulose fibres. This pioneering study, along with other subsequent works, led to the discovery of a new nanomaterial called nanocrystalline cellulose (CNC) [38]. Since then, many works concerning the characteristics and the processing of nanocellulose have been published [39,40,41,42,43].

Isogai et al. [44] extracted cellulose nanofibers (CNFs) from wood pulp. Isogai’s method produces nanofibers exploiting TEMPO-mediated oxidation. These fibres have diameters that are from 5 to 60 nm long and lengths of many microns.

CNC has been derived by treating cellulose pulp with strong acids. In this way, the unstructured aggregates are dissolved, whereas the crystalline parts of the fibrils remain intact [45]. The extraction product, CNC, has a diameter of 5 nm and 20–100 nm length. The source of cellulose and extraction procedures influence the dimensions of both CNFs and CNCs.

Peyre et al. [46] produced cellulose nanocrystals (fine fraction) and two different-length-scale microparticles (medium and coarse fraction).

The method consists of the oxidation of microgranular cellulose by TEMPO oxidants. A pH = 8 environment and a bromide-free route were applied. The finest fraction was obtained by the centrifugation and separation of the supernatant. An ordinary filtration allowed the middle fraction (smaller microcrystals in micrometre-length scales) to be separated from the coarse fraction.

BNC does not have the same quantity of impurities that lignin and hemicellulose present. Compared to the nanocellulose obtained from wood [47], it has a higher crystallinity (80%). It is secreted by bacterial fermentation of a glucose-based culture medium [48]. The bacteria strains involved are *Agrobacterium*, *Salmonella*, *Aerobacter*, *Escherichia*, *Sarcina Rhodobactor*, *Komagataeibacter* and *Gluconacetobacter xylinus* [49,50,51]. A washing procedure is applied to bacterial cellulose (NaOH 1 M). In this way, BNC is extracted and the unwanted residues of proteins are removed. BNC can form hydrogels independently of the source of nanocellulose fibrils, which have an average diameter of 100 nm and length of the order of micrometres.

### 3.3. Cellulose Solubilization

The main limitations of cellulose are related to its dissolution in water and more organic solvents. Researchers are being forced to use other solvent systems for cellulose dissolution, such as ionic liquids (ILs), NaOH/urea and NaOH/thiourea. ILs are handled with some complication and difficulty. By far the most widely used solvent system is NaOH/urea because it is easily obtainable, inexpensive and increases the dissolution rate of cellulose at low temperatures [52]. To improve the solubilization of poorly soluble natural polysaccharides (such as cellulose, chitin and chitosan) in alkaline urea systems, a considerable amount of work has been performed since 2008, also exploiting their derivatization. Quaternary ammonium cellulose (QAC) with a degree of substitution (SD) of 0.20–0.63 was homogeneously produced, for the first time, by the reaction of cellulose with trimethylammonium chloride (CHPTAC) in an aqueous solution of NaOH and urea [53]. This work confirms alkali/urea solvent as a homogeneous reaction system for the chemical modification of poorly soluble polysaccharides, giving indications on the path for future work.

### 3.4. Crosslinking

Cellulose gels, depending on their crosslinking density, form a network with meshes of different diameter and stiffness [54,55,56,57]. Figure 5 illustrates some of the interactions which are at the basis of the formation of cellulose hydrogel.

Thanks to properties such as biodegradability, non-toxicity, flexibility and biocompatibility, nanocellulose hydrogels are employed in multiple biomedical applications.

On the other hand, to promote crosslinking and to produce hydrogels, it is necessary to ensure the efficient dissolution of cellulose; this involves the breaking of both intra- and intermolecular hydrogen bonds [51]. Several chemical protocols have been studied to dissolve cellulose in water, organic solvents or IL [59]. Cai et al. developed a procedure in aqueous solutions of NaOH/urea and LiOH/urea to dissolve cellulose at temperatures as low as −10 °C [60]. The best results were obtained with urea at 12% concentration. After dissolution, an optically transparent hydrogel crosslinked by epichlorohydrin crosslinker (ECH) was formed at −3 °C. 

The hydrogel’s preparation determines its network structure. The swelling of cellulose fibres depends on ion diffusion from the solution toward the gel centre [61]. Swelling power is an important parameter because it allows not only the control of the pore size and its distribution, but also the gel’s mechanical properties. It is important to control the pore size of the gel because it is connected to the diffusion of biomolecules within the matrix [60]. If the mesh size is smaller than the diameter of the biomolecule, the diffusion of biomolecules is blocked by filtration.

Hydrogels usually appear translucid; more turbidity is evident when there is a higher solid content, and they become coloured if additives such as ions, proteins and polymers are present. The final application can guide the choice of the most appropriate technique for processing the hydrogel. The pH and the ionic strength of the aqueous solution can be tuned, and additives (salts, sugars and amino acids) can be introduced inside the gel to balance the osmolality. The cellulose backbone can be functionalized with substituent groups (such as methyl, carboxylate or amine) and modified to trap biomolecules (enzymes, antibodies and peptides) [62,63,64,65].

Cellulose derivatives, such as methylcellulose and carboxymethyl cellulose are water-soluble; they have been widely applied for the preparation of nanocomposite hydrogels as drug carriers, promoters of wound healing and other pharmaceutical applications.

Seera et al. [66] evaluated the properties of hydrogels obtained by microcrystalline cellulose (CMC) and polyvinyl alcohol (PVA), comparing the properties of physically crosslinked DMSO-NaOH/urea systems with those of hydrogels chemically crosslinked in water-NaOH/urea, and concluding that the latter had better rheological properties. Rheological analyses showed that CMC underwent a successful crosslinking with PVA, evidencing that storage modulus dominated the loss modulus until the crossover point was achieved. Drugs and water are loaded thanks to the porous structure. The swelling studies and the rheological results are linked. In fact, when crosslinking level lowers, the swelling level increases and vice versa. Release studies performed with 5-fluorouracil have shown promising results for the use of chemical crosslinking of PVA with CMC to overcome the water solubility issues presented by pure microcrystalline cellulose, which hinder its use in drug delivery.

Peng et al. [67] produced a cellulose/clay hydrogel by crosslinking cellulose, carboxymethyl cellulose and clay nanosheets with epichlorohydrin in a NaOH/urea aqueous solution. The clays (epoxidized montmorillonite) were homogeneously dispersed in the cellulose matrix. This hydrogel incorporated with the intercalated clays exhibited better absorbent and mechanical properties compared to the hydrogel containing the unmodified clay.

Jun Liu et al. proposed two strategies to produce hydrogels: pre-adsorption and in situ adsorption of hemicellulose. The first method is performed in two steps: (a) hemicellulose is adsorbed onto CNF and the film is obtained using hot mixing; (b) the film swells during immersion in water at room temperature for a day. The second method is a process in which the adsorption of hemicellulose and addition of water (for swelling) occur simultaneously [63,68,69]. The highest adsorption capacity on CNF was governed by the preadsorption method using xyloglucan as a crosslinker.

Among the different types of cellulose, bacterial nanocellulose is the most promising and suitable biomaterial [47,48,49,50,70] for producing hydrogels. This is because BNC is produced as a 3D nanostructure with high tensile strength and elastic modulus [47,49]. Moreover, BNC-based hydrogels still maintain this network structure at a high level of hydration. Even if BNC has production costs that are more disadvantageous than cellulose nanofibers (CNF) and cellulose nanocrystals (CNC), several studies reported that its hydrogels can be applied for the controlled and targeted delivery of antibodies, enzymes and drugs [54].

### 3.5. Grafting

Graft polymerization is a technique used to improve the physical and chemical characteristics of natural fibres, introducing some modifications. The method chosen to obtain grafting is radical polymerization [65,71,72]. This method does not allow complete control of the reaction process. As a result, the molecular weight of the product is widely distributed, and the grafting chains have variable lengths [73]. To solve these problems, reversible-deactivation radical polymerization is applied. Anin et al. made and characterized a hydrogel of bacterial-derived nanocellulose and acrylic acid (AA) for drug delivery. Acrylic acid monomers were implanted onto cellulose fibres, and the subsequent irradiation by electron beams accelerated the gelation. This hydrogel is pH- and temperature-sensitive due to the presence of acrylic acid [74]. The water absorption capacity was evaluated, and it was observed that BNC-(AA) hydrogels at pH 7 reach swelling after 48 h. At pH 10, equilibrium is reached in 24 h. The structure that is created enables this swelling mechanism, and, in particular, image analysis has evidenced that acrylic acid influences the pore size of the gel.

The gel porosity is important because it determines the kinetics of the drug release. Badshah et al. published a work concerning BNC matrices loaded with two different drugs: famotidine and tizanidine. The first drug is slightly soluble in water; the second is very soluble. The loading inside the matrixes of the two drugs was uniform. The release percentage in 15 min was 80%. Drugs underwent a faster release thanks to the porosity of BC network and its hydrophilicity [55].

The Diels–Alder reaction is promising for the design of advanced materials from cellulose nanocrystals (CNCs). It involves the accurate grafting of reactive Diels–Alder moieties under various conditions without affecting the nanocrystal structure. Garcìa-Astrain et al. [75] performed a grafting of maleimide to the surface of CNCs. Then, they employed the Diels–Alder reaction to crosslink functionalized CNCs with furan-modified gelatine. The hydrogel was stabilized by a second crosslinking performed by coupling on the amide between CS and furane-modified gelatine, obtaining a completely renewable bio-nanocomposite formulation. The authors studied the role of CNCs as a stabilizer and their effect on the swelling and viscoelastic properties.

### 3.6. Cellulose Hydrogels with Metal Nanoparticles or Cations

Gulsonbi et al. [76] used a green chemical synthesis process for the preparation of a biodegradable semi-interpenetrating nanocomposite hydrogel of silver/carboxymethyl cellulose-poly(acrylamide). The poly(acrylamide)/carboxymethyl cellulose hydrogel network was obtained by performing a fast-redox polymerization of N, N-methylenebisacrylamide (MBA) with CMC. The silver nanoparticles AgNPs within the hydrogel network were successively synthesized by in situ bioreduction of AgNO_3_ using *Azadirachta indica* (neem) plant extract under atmospheric conditions.

In 2018, Basu et al. [77] proposed the setup of a Ca^2+^ crosslinked wood-based nanofibrillated cellulose (CNF) hydrogel, thereby achieving increased knowledge on the use of nanocellulose aimed for wound healing. The obtained hydrogel, through a soaking procedure, can carry proteins of various sizes and isoelectric points. The release of the proteins was supervised, and their determining parameters were evaluated. The results showed that the electrostatic interactions between proteins and the structure of the negatively charged CNF hydrogel play a central role in the loading process, and that protein release is controlled by Fickian diffusion. An increase in protein size, as well as the presence of the positive protein charge, induced a slower release process. Positively charged proteins have also been found to improve the stability of the hydrogel. The hydrogel composed of CNF and the crosslinking agent Ca^2+^ has a structure that allows the transport of proteins and their release, maintaining their structural stability without compromising their activity. It is expected that by using the charge-adjustable properties of the hydrogel, CNF release profiles can be tailored for specific therapeutic needs.

F. Lin et al. [78] reported an easy strategy for the development of a hydrogel drug delivery system, employing a simple addition of Fe_3_O_4_ nanoparticles to a mixed β-cyclodextrin (β-CD)/cellulose solution. By dispersing this dropwise within a coagulant bath of CaCl_2_, magnetic nanocellulose nanoparticles were obtained. It has been hypothesized that β-CD grafting confers high drug-loading capacity to the hydrogel. The incorporation of Fe_3_O_4_ nanoparticles, through the fast and reversible deformation of a 3D network subjected to an external magnetic field EMF, can improve the drug delivery modes. Thus, through an environmentally sustainable methodology, it was possible to design a smart cellulose hydrogel that combines rapid response and remote control of remote drug release under the effect of EMFs by switching the “on-off” mode.

## 4. Chitosan-Based Hydrogels

Hydrogels based on chitosan are diffused and, in recent decades, have received growing interest for the great possibilities that they offer in pharmaceutical applications. Chitosan is obtained from chitin, an abundant naturally occurring polymer, but chitin, compared to cellulose, the other abundant natural polymer, has fewer applications. This is for several reasons, especially for the poor solubility properties of this polysaccharide. Nowadays, much research work is being carried out to overcome this hurdle because the advantages of using chitin and its derivatives [79] are numerous, and due not only to its natural abundance, but also to its renewability, biodegradability and non-toxicity.

Chitin is characterized by the presence in its structure of the major ordered crystalline microfibrils α and β [80]. The α form is more frequent and stable; it is found in fungi, shrimp and insects. Spines, annelids and squid pens [81] are the main source of β-structure chitin. Its strong intra- and intermolecular hydrogen bonds make chitin extremely aggregated and cause its insolubility in common solvents [82].

For this reason, chitin is converted into chitosan by extensive alkaline deacetylation. This polysaccharide is formed by units of N-acetyl-D-glucosamine and D-glucosamine β(1→4) linked. In the solid state, this biopolymer is semi-crystalline with a wide range of polymorphs depending on formation conditions [83] (Figure 6).

### 4.1. Chitin Deacetylation

The main method of deacetylating chitin consists of a treatment with highly concentrated (40–50%) NaOH solution for 6 h at 107 °C [15]. The treatment with concentrated alkali has a large environmental impact and high cost, and a large amount of water is needed in the neutralization stage. All these elements limit the use of chitosan in industry.

Therefore, it has been fundamental to find new environmentally and economically sustainable alternatives. An enzymatic method has been proposed. This procedure involves deacetylation by the Chitin Deacetylase (CD) enzyme produced by fungi or bacteria [84]. CD catalyses the hydrolysis of N-acetamido bonds. The method is also eco-friendly due to its mild reaction conditions. Kim et al. [85] obtained chitosan at 60 °C and pH 5.5 using extracellular CD. However, this enzymatic method is not suitable for industrial use because of the high cost of CD and long reaction time [86]. Liu et al. [87] developed an alternative green process by using glycerol as the reaction solvent at 180 °C, thereby lowering the NaOH concentration needed for the deacetylation reaction. Glycerol is a recyclable green solvent obtainable as a by-product of biodiesel. The idea of using it as a solvent within the chitosan production procedure is a perfect example of circular economy and by-product recycling. In fact, glycerol and NaOH can be recovered and reused for another deacetylation reaction. The optimal procedure involves treating chitin with 30% NaOH and glycerol, keeping a 1:40 chitin/glycerol ratio. During the process, 1% of water must be added; otherwise, the polymerization of glycerol in an alkaline environment occurs.

### 4.2. Derivatization of Chitosan

Owing to its biodegradability [88], biocompatibility, adhesion and properties, chitosan is of great importance in various applications, such as the food industry, pharmacy, nano-biotechnology, etc. [89,90,91,92,93,94,95]. In addition, chitosan possesses antibacterial and antioxidant action because of its intrinsic positive charge [72], and therefore is capable of crossing the cell membranes of negatively charged bacteria [96,97].

The modification of natural polymers is one of the main topics in bio-technology research because it allows new derivatives with unique properties to be developed. Polysaccharides can be modified using several methods. Chitosan can undergo chemical modification at free amino groups from the deacetylated units at the C-2 position and hydroxyl groups at the C-3 and C-6 positions [98]. Chitosan is functionalized to improve certain characteristics, such as solubility, hydrophilic character and affinity towards bioactive molecules [55,71]. Understanding the interdependence between chemical structure and properties is important for the design of innovative materials. In recent years, a click chemistry approach, grafting on copolymerization, coupling with cyclodextrins and ionic liquid reactions have been explored for the development of new derivatization strategies.

### 4.3. Derivatization through Ionic Liquids (ILs)

Nowadays, the derivatization of chitin and chitosan are easily performed in ionic liquids (ILs), which are a versatile medium. Ionic liquids are salts that remain liquid below 100 °C. The major advantage in the use of ILs is their recycling, which is important for their use in industrial-scale applications. Many ILs have been considered “green” solvents because of their low vapor pressure, nonflammability and high solvation potential [99,100]. Chitosan-IL solutions are particularly useful for derivatization reactions because this reaction does not occur in water [101,102].

### 4.4. Click Reactions

Click reactions are quick, modular and high-yield reactions; their modularity resembles that of natural processes. They can be conducted in a wide range of pH (4–12), under mild temperature conditions (25–37 °C) and are insensitive to water and oxygen. The regioselective products that are generated do not need complicated purification processes.

Click chemistry has also been found to have important applications in cell-based drug delivery. Mesenchymal stem cells (MSCs) have been studied as carriers of anticancer drugs by D.G. Ackova et al. [103] because of their attraction to tumours. S. Lee et al. modified MSC surfaces by combining metabolic glycol engineering and copper-free click chemistry (SP-AAC: strain-promoted azide-alkyne cycloaddition). The authors functionalized MSC surfaces with CNPs (chitosan NPs), monitoring in the long term [104]. The successful application of click chemistry combined with glycoengineering to tumour cells shows that they can be functionalized with receptors for a specific anticancer drug [105], and moreover immunostimulatory molecules can be expressed [106].

### 4.5. Cycloadditions

Cycloadditions are the most widely used click reactions to functionalize chitosan. The Huisgen reaction is a cycloaddition between alkynes and azides, giving two tiazole regioisomers [107,108]. It can be conducted with or without Cu(I), which is the catalyst. It is one of the most examined chemoselective “click” reactions, which takes place in an aqueous environment at room temperature. The Diels–Alder reaction is a [4 + 2] cycloaddition between a diene and a dienophile, producing molecules with an unsaturated six-membered ring. This is a central click reaction which is thermodynamically reversible [109,110]. Diels–Alder cycloaddition, compared to Huisgen cycloaddition, is advantageous because it allows the production of materials that can vary their physicochemical properties with temperature. Furan, because of its dienic behaviour, is a molecule that offers many opportunities to study the potential of Diels–Alder cycloaddition between furan–chitosan derivatives and dienophiles such as maleimides. This method has enabled the realization of a new chitosan hydrogel with interesting drug transport characteristics, suitable for the development of new biotechnological and biomedical materials. By using renewable resources, such as furfural, and simple experimental conditions, new chitosan derivatives can be obtained. FT-IR and NMR analysis of N-(furfural) chitosan suggested the possibility of producing derivatives with different degrees of furfural substitution. Therefore, one strategy would be to study the chemical interactions between furfural chitosan derivatives and maleimide compounds by Diels–Alder reactions. Hydrogel networks obtained using the reaction of N-(furfural) chitosan with bis-maleimide have remarkable viscoelastic properties; moreover, they exhibit interesting and controlled release properties appropriate for the improvement of biological and biomedical applications. Green synthesis has been proposed to produce biocompatible microgels [111]. The microgels were chosen as possible carriers of drugs such as vancomycin hydrochloride (VM) to be administered at the colonic level. Chitosan and dextran were chosen as bio-based building blocks, and, respectively, modified by employing Copper (II) as a catalyst with alkyne and azide groups and crosslinked in W/O miniemulsions, obtaining microgels with a size and crosslinking density that can be adjusted. In addition, such microgels are degradable by dextranase, an enzyme found in the colon [111], or by hydrolysis occurring in an alkaline environment [112]. The degree of crosslinking of microgels can be easily adjusted by the azide/alkyl molar ratio in prefunctionalized biopolymers. Microgels have the ability to release the drug in the colon region under the presence of dextranase and to protect it in the stomach (pH 2–4) or intestine (pH 6–7.5).

The conditions that must be imposed for the mentioned reactions to take place are not always mild and eco-friendly. The usage of solvents and catalysts that can harm the environment is strongly discouraged by the guidelines of Green Chemistry. For these reasons, research is being oriented toward the development of eco-friendly solvents.

### 4.6. Grafting

ILs have important solvent capacity and have been used to obtain grafting on chitin or chitosan. Methacryloyloxyethyl trimethylammonium brushes were formed on chitosan by single electron transfer radical polymerization in 1-butyl-3-methylimidazolium (Bmim) Cl [113]. The synthesis of chitosan graft polyethylenimine copolymers was realized in BmimAc [114].

Ionic liquids have the ability to realize inter-polysaccharide reactions that, under other conditions and media, have been carried out with great difficulty. With the use of a mixture of two ILs, AmimCl as the solvent and 1-sulfobutyl-3-methylimidazolium hydrogen sulfate (SmimHSO4) IL as the catalyst for the reaction, it was thus possible to realize chitosan graft oxycellulose [115].

Graft polymers have also been obtained through eco-friendly procedures involving neither Cu nor IL. As an example, we can consider the work of Y. Chen et al. [116], in which the successful grafting of carboxymethyl chitosan (CMCS) with Marine Collagen Peptides (MCPs) in aqueous phase using ethyl-(dimethylaminopropyl) carbodiimide (EDC)-mediated coupling method is presented. MCPs from the skin of *Nile tilapia* (*Oreochromis niloticus*) were grafted onto the CMCS chains and the CMCS-MCP sponges were prepared by freeze–thaw cycling and freeze-drying. In vivo and in vitro haemostatic experiments demonstrated that CMCS-MCP sponges have brilliant haemostasis activities, which could be attributable to the synergistic effect of CMCS and MCPs through multiple blood coagulation approaches. The eco-friendly process was driven in various steps. The solutions of CMCS and MCP were separately prepared. (EDC) and N-hydroxy succinimide (NHS) were added step-by-step to the CMCS solution. The obtained CMCS-EDC-NHS solution and MCP solution were mixed and continuously stirred. Then, dialysis in distilled water for three days was performed, periodically changing the distilled water. The dialyzed solution was then treated by freeze–thaw cycling and the dried samples were subjected to freeze-drying.

In a recent study [117], Logigan et al., according to a protocol reported by Han et al. [118], performed a Michael addition of poly (ethylene glycol) methyl ether acrylate (PEGA) to the amine groups and then prepared chitosan particulate hydrogels (CPHs). Therefore, chitosan underwent a double crosslinking, ionic and covalent, in a water/oil emulsion. The studied process parameters were polymer concentration, stirring speed and quantity of ionic crosslinker. The characterization studies performed through infrared spectroscopy, scanning electron microscopy, light scattering granulometry and zeta potential showed that modified chitosan, compared to neat chitosan, has better control of dimensional properties and morphology. Swelling behaviour in acidic and neutral pH conditions maintained a pH dependence after the procedures of grafting and crosslinking. The prepared materials were tested by studies concerning the in vitro delivery of levofloxacin (LEV). The tests gave satisfying results.

A chemically modified chitosan injectable hydrogel for controlled drug delivery was developed [119] by A.K. Mahanta et al., who grafted ester-diol-based polyurethane onto highly hydrophilic chitosan, which turned into hydrogel through hydrophilic–hydrophobic equilibrium. The grafted copolymer hydrogel was made in a dilute acetic acid medium and exhibited a higher contact angle and less swelling compared to the pure hydrogel. Presenting an interconnected porous network, it appears suitable to be employed for the release of drugs. Antibacterial tetracycline hydrochloride and anticancer doxorubicin hydrochloride were employed to study the release of drugs in various fields of application. The biocompatibility of the hydrogel was tested using melanoma cell line B16-F10, and the tests showed that such a hydrogel performed better than pure chitosan. The authors observed that the copolymer hydrogel offered better cellular uptake compared to the pure drug. An in vivo gelling study in an animal model verified that the graft copolymer could be used as an injectable hydrogel for disease control.

### 4.7. Beads and Coated Beads

Ionic liquids have been recently applied in green processes that produce porous microbeads suitable to be employed as drug delivery systems [120]. Beads are spherical material particles (in the case of microbeads, typical sizes range from 5 microns to 1 mm) largely used in biomedical science and in cosmetics as exfoliating agents [121].

Chitosan-derived biomaterials can form hydrogel beads for oral drug delivery. Because of their poor solubility under physiological conditions, chitosan hydrogel beads become swollen. Additionally, these beads are highly biocompatible, and this property allows drugs to disperse more easily. The entrapment of drugs occurs thanks to the mesh structure produced by chitosan chains crosslinking [122]. An excellent strategy to stabilize nanoparticles is the creation of a surface coating. As a coating material, Silk Fibroine (SF) appears to offer great mechanical stability and act as a barrier to the diffusion of the encapsulated drugs, which can be attached to the available site for functionalization. Moreover, it can be easily processed and formulated in water and allows the drug release to be significantly slowed [123,124,125,126] thanks to the control of the coating thickness.

Song et al. [127] proposed an alternative method for cellulose coating of chitosan hydrogel (CS) beads, useful as drug carriers. The beads were coated with cellulose dissolved in the ionic liquid (IL) 1-ethyl-3-methylimidazolium acetate. When water molecules entrapped in the CS beads came into contact with the mixture of cellulose and IL, they diffused outward. The regenerated cellulose formed a coating on the beads’ surface. The model drug verapamil hydrochloride (VRP) was inserted into the coated beads. When tested in simulated gastric fluid (pH 1.2), the impregnated beads released VRP and thus were proven to be useful as carriers for controlled drug release. A sustainable alternative to chitosan has been developed by Rogers et al., who realized chitin microbeads using ionic liquid1-ethyl-3-methylimidazolium acetate ([C2mim]-[OAc]), which not only extracts the chitin from waste shrimp shells but also produces porous microbeads by a coagulation process deploying eco-friendly polyethylene glycol (PEG) [121]. This alternative chitin beads, produced by using chitin with high molecular weight, possesses a homogeneous shape and a uniform size, not obtainable from commercial-grade chitin.

### 4.8. Radiation Synthesis

Radiation synthesis is a relatively new technology that is effective and environmentally friendly. Owing to the high temperatures that are produced in a relatively short time and that quicken reactions more than traditional thermal conditions, microwave irradiation has been exploited to create hydrogels [128,129]. Ultrasound-assisted irradiation has been found to be useful for the dispersion of monomer droplets, the generation of free radicals [130,131] and the control of the molecular weight and swelling ratio [132]. Li et al. demonstrated that ultrasonic waves could loosen the agglutination of collagen in water and increase yields in the extraction of collagen from bovine tendon enzyme [133].

### 4.9. Metal–Hydrogel Hybrid Nanoparticles (NPs)

To improve the encapsulation of the drug selected for a particular cancer treatment, doxorubicin, NPs were obtained [98] by mixing chitosan and graphite oxide in an aqueous solution and acetic acid and then adding rectorite [134].

Low-molecular-weight chitosan (LMWC) stabilized with gold nanoparticles was synthesized [120]. The system was loaded with the model chemotherapeutic drug doxorubicin (DOX). Then, the coating of the nanoparticles with folic acid (FA) and fluorescein (FL) was performed.

In a recent study (2022), M. Chelu et al. [73] proposed an easy, green synthesis methodology to produce zinc oxide nanoparticles using three gums (Acacia gum, Guar gum and Xanthan gum) of biological origin. Then, zinc oxide nanoparticles were loaded into a chitosan hydrogel functionalized with propolis extract. This study demonstrated that polysaccharide gums exhibit a chelating behaviour and that this approach seems suitable for the synthesis of ZnO nanoparticles with controlled morphology. The incorporation of various molecules into the hydrogel allowed composites to be obtained that were cytocompatible in L929 fibroblast cell culture, in a range of concentrations between 50 and 1000 μg/mL, as proven by MTT, LDH and Live/Dead double-staining assays. The composite green hydrogels exhibited better performance in a wide concentration range for the future development of platforms as effective drug delivery systems.

An important field of application of hydrogel/metal complexes is wound dressing; in fact, the poor mechanical properties of chitosan hydrogels limit their usage for wound healing. To overcome this drawback, Yajuan Xie et al. [135] adopted a LiOH/urea solvent system to synthesize chitosan hydrogel as an alternative to the traditional glacial acetic acid method. With the purpose of improving the mechanical and antibacterial properties of chitosan hydrogels, Ag nanoparticles were introduced into the chitosan hydrogel networks. The synthesized hydrogels were characterized by FTIR, XRD and TEM. Swelling, mechanical and antibacterial properties and wound-healing efficacy were estimated. The porous 3D structure of the new hydrogel was confirmed by the results. This network retained structural integrity even with strain exceeding 90%. Additionally, the hydrogel exhibited better antibacterial performance than the controls, alongside increased collagen deposition, promoting wound healing. Therefore, the synthesized hydrogel appears suitable for applications in the fields of biomedicine.

## 5. Hyaluronic-Acid-Based Hydrogels

Hyaluronic acid (HYA)-based hydrogels have been developed and studied in recent years for biomedical applications such as tissue regeneration, drug delivery, gene therapy, diagnostics, etc. The reasons why researchers have been investigating this polymer for the realization of hydrogels are related to its chemical–physical properties. HYA is a naturally linear polysaccharide consisting of alternating units of d-glucuronic acid and N-acetyl-d-glucosamine, linked by β-1,3- and β-1,4-glycoside bonds (Figure 7).

In addition to performing important functions in the extracellular environment of human body’s tissues, helping to maintain their chemical balance and mechanical integrity, it also has important functions in the intracellular matrix. It binds cell receptors and therefore regulates cellular processes such as adhesion, proliferation and, as a consequence of inflammation, the healing of wounds, tissue regeneration, tumour development and metastasis [136,137,138].

The hydrophobicity and biological activity of HYA hydrogels can be controlled by introducing chemical modifications inside HYA molecules [138,139].

### 5.1. Crosslinking Methods to Create HYA-Based Drug Delivery Systems

The chemical modifications of HYA involve three functional groups: the carboxyl group of glucuronic acid, the primary and secondary hydroxyl groups and the N-acetyl group. Specifically, carboxylates can be modified by esterification and carbodiimide-mediated amidation reactions; hydroxyls can be modified by etherification and esterification.

It is possible to create HYA hydrogels by esterification with targeting molecules, and particularly molecules whose receptor is found to be most highly expressed on the surface of the cancer cells towards which the oncotherapy active ingredients are to be delivered. For example, in the case of ovarian carcinoma, the folic acid receptor FR-α is overexpressed, which allows folic acid to function as a targeting molecule and deliver the active ingredients (cis-platinum, doxorubicin) directly to the tumour cells, without cytotoxic effects on healthy cells. FR-α is also overexpressed on the surface of activated macrophages [140], which cause autoimmune inflammatory diseases, such as rheumatoid arthritis. Therefore, hydrogels formed by hyaluronic acid esters and folic acid can be used for the release of anti-arthritic drugs (methotrexate).

Another molecule of interest for producing targeting hydrogels, and on which research has been focused, is tyramine, whose TAAR1 receptor [141] is overexpressed in some tumours, such as liver cancer. Hydrogels consisting of hyaluronic acid and tyramine, after their preparation, can incorporate molecules such as interferon-α2a for therapy against this type of tumour [142]. The realization of the hydrogel can be achieved through the formation of a HYA-Tyramine conjugate, through carbodiimide-mediated coupling of the amine to the carboxyl of the glucuronic unit [143]. After the realization of this coupling, further crosslinking reactions can be carried out [144,145].

E. Larraneta et al. [146] developed a method to realize the crosslinking reaction between HYA and the crosslinker in solid phase, thus producing HYA-based biomaterials with a defined form. The procedure is environmentally friendly because it is carried out without any organic solvents or potentially toxic substances. It was applied to an aqueous mixture containing 5% (*w*/*w*) of HYA and different concentrations of Gantrez S97 (GAN) (1, 3, 5% *w*/*w*). GAN is the acid form of the copolymer formed by methylvinylether and maleic anhydride; it includes acid groups suitable to react with the hydroxyl groups present in HYA chains and to form esters. The reaction blends were dried, and after heating the solid material at 80 °C for 24 h, hydrogels were obtained. HYA/GAN systems exhibited affinity for the MB (Methylene Blue) molecule and loaded up to 0.35 mg of MB per mg of hydrogel. Moreover, they sustained the MB release over two days and presented anti-infective properties.

The inverse electron demand Diels–Alder (IEDDA) click reaction between tetrazine (Tz) and norbornene (Nb) [147] can occur without the need for any catalyst. This reaction can be applied to realize drug delivery systems because it produces only small amounts of nitrogen gas and no other toxic side products. The IEDDA click reaction was applied by Jo et al. [148] to formulate a stimuli-responsive HYA-based hydrogel for the controlled release of doxorubicina (DOX), with a crosslinker containing diselenide (DSe) bonded to ditetrazine (DTz). HYA was first conjugated with norbornene groups and then reacted with a DSe-DTz crosslinker. The performance of the hydrogel was found to depend on the various crosslinking ratios, whereas the drug release study of hydrogels showed that NIR irradiation rapidly de-crosslinked to the hydrogel linear chains. A burst release of DOX was found to occur at the beginning of the study and a sustained DOX release afterwards. The anti-tumour activity of DOX-loaded hydrogels was found to be similar to that of the free DOX.

### 5.2. Crosslinking Methods to Create Tissue Engineering Systems

The HYA-tyramine hydrogel also has the potential to be used, using enzymatic methodologies [149], for the fabrication of a composite hydrogel with Silk Fibroin (SF), a protein that has proven particularly useful in systems designed to incorporate cell cultures for delivery into tissues affected by degenerative diseases. In the case of arthritic disease, chondrocytes will obviously be used. Rather than performing chemical crosslinking by photo-initiator or solvent-mediated organic chemistry reactions, the use of enzymatic crosslinking (HRP enzyme, horseradish peroxidase in H_2_O_2_) was preferred, thus avoiding the toxicity of the reagents and aggressive conditions for the cells. This methodology has shown minimal cytotoxicity with the in situ gelation of HYA hydrogels. Enzymatic crosslinking of HYA/SF composite hydrogels can modulate the viscoelasticity and stiffness in relation to the concentration of HYA-Ty and the HYA/SF polymer ratio. This allows a better control of gelation and degradation to be achieved, providing a more versatile and reliable platform for drug delivery than simple HYA-Ty hydrogels [150].

## 6. Alginate-Based Hydrogels

The sodium alginate (SA) polysaccharide is obtained from brown seaweed (Figure 8). It has been found to be useful for drug release, tissue engineering and encapsulation of proteins.

### 6.1. Crosslinking to Other Biopolymers [151]

SA hydrogels have several drawbacks, including low elasticity and poor cell adhesiveness [152]. Moreover, the ionic crosslinking of SA hydrogels through Ca^2+^ quickens the gel process, but it is not advantageous for producing a uniform drug distribution. Another drawback is that SA hydrogels crosslinked by divalent cations tend to dissolve if the environment contains monovalent ions, since the divalent cations are replaced by monovalent ions [153]. To overcome these difficulties and to improve the performance of SA hydrogels, attempts have been made to link SA to other polymers with satisfactory properties. One of these polymers is Silk Fibroin (SF), a major component of silk. It has biological activities, interesting mechanical properties, biocompatibility, easy manipulation and degradability [154,155,156,157].

**Figure 8 molecules-28-02107-f008:**
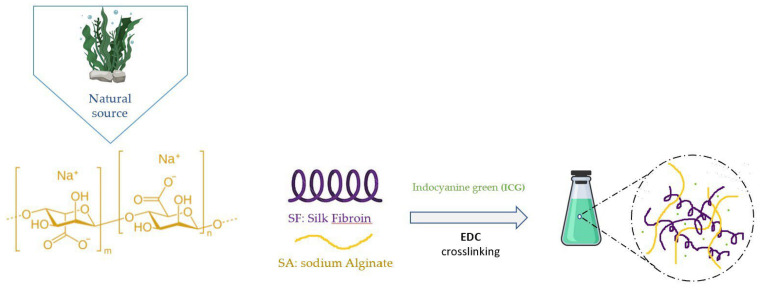
Alginate molecule and the synthesis of hydrogel by the addition of Silk Fibroin and crosslink1-ethyl-3-(3-dimethylaminopropyl) carbodiimide (EDC) [157].

Hydrogels with SF are more similar to extracellular matrices (ECMs), and by using 1-ethyl-3-(3-dimethylaminopropyl) carbodiimide (EDC) as a crosslinking agent, SF/SA gels form faster than SF gels or SA gels alone [158]. They have good biocompatibility and are an NIR-responsive biomaterial, so they offer photothermal-modulated drug release when the NIR light is turned on. Cong et al. developed an alginate hydrogel/chitosan micelle pH-sensitive system. It was found to be promising for the sustained release of hydrophobic drugs [159]. After the encapsulation of Emodin (EMO) micelles, the sodium alginate hydrogel showed sensitiveness to pH and was demonstrated to be suitable for oral drug delivery. The investigation of its physical characteristics in simulated digestive fluids led to the discovery that micelle/hydrogel ratio affects swelling, drug release in vitro and biodegradation. For the ratio, 1:1 sustained release was obtained, whereas the ratio 1:3 characterized a profile specific for the colon. Several mechanisms occurring simultaneously, or following a complex process, were found to be involved in the release of drugs from these two formulations.

### 6.2. Nano- and Microparticles

AgNPs are manufactured by various methods, including chemical, physical, and green ones [160,161,162,163]. Chemical and physical methods do not have an accomplishable economic profile (expensive, low production rates) and are not environmentally sustainable owing to the toxicity of the chemical agents used [164]. Green chemistry methods are preferred because the reagents of the synthesis are plant extracts or microorganisms, such as fungi and bacteria [160,165,166,167]. To increase particle stability, and to prevent their aggregation, various coating agents have been employed. Polyethylene glycols (PEGs), due to their stability, hydrophilicity and non-irritating action on the skin, have been frequently used [168,169]. If the particles of AgNPs are suspended in a hydrogel matrix and not left as free particles in water, their topical application for the treatment of skin infections seems to be more effective. Several gelling agents have been reported in the literature for the preparation of antimicrobial hydrogels. Some of them enhance the efficacy of the loaded antimicrobial agents [166,170] or facilitate their topical application, such as sodium alginate [171].

AgNPs were obtained through biological synthesis, using an economical and environmentally friendly methodology, which exploited Fusarium verticilloides fungus. The activity of AgNPs was studied by noting the effect of coating agents, including PEG 6000 sodium dodecyl sulfate (SDS) and β-cyclodextrin (β-CD), on the stability and antimicrobial activity of AgNPs. The parameters used were the viscosity, spreadability and antibacterial and drug delivery in vitro activities. The efficacy of the hydrogel containing 0.1 percent AgNPs was compared with that of commercially available silver sulfadiazine 1 percent cream (Dermazin).

The performances of the hydrogel formulations were superior to those of the silver sulfadiazine cream as far as the ability to heal wounds and antimicrobial proprieties were concerned.

Nanomaterials based on hydroxyapatite (HA) have attracted the attention of researchers because their mechanical and biocompatibility properties allow them to solve problems related to the filling or coating of hard tissues. Moreover, HA has been employed for the release of several anti-inflammatory and antibiotic drugs [172,173,174,175], but its usage as a carrier [176,177,178,179,180] is expected to be very promising for other pharmaceuticals. In the work of Rial et al. [181], these properties were exploited for the development of a hybrid hydrogel system obtained by combining external and internal gelation. The external gelation was obtained using two external baths, whereas the internal gelation was caused by the bonds formed between Ca^2+^ presented on the HA surface and the oxygen sites of COO– groups of alginates [180]. HA was synthesized according to a procedure described in [182] and preliminarily doped with Propranolol hydrochloride and cloxacillin sodium salt monohydrate used as drugs. Using this methodology based on the use of a microfluidic system, Calcium-Alginate microparticles (Ca-ALG) and core–shell Alginate-Chitosan microspheres (Ca-ALG-CHI) were synthesized with the presence of drug-doped Hydroxyapatite (HA) in their inner matrix. The methodology allowed crosslinked hydrogel microparticles (HMPs) with homogeneous sizes and morphologies to be obtained, integrating external and internal gelation.

The biocompatible nature, the porosity and the high drug-loading capacity of mesoporous silica nanoparticles (MSNs) have prompted researchers to explore their potential in drug delivery for the treatment of various diseases [183]. The pore size of MSNs determines their usage in drug delivery systems. SBA-15, belonging to the SBA (Santa Barbara Amorphous) family, have pore sizes of 5–15 nm. Liao et al. reported that their targeted receptor was overexpressed on cancer cells [184].

Thus, the MSN-Alg microspheres are suitable for use in pharmaceutical formulations for their efficiency in transmembrane controlled release of cell-membrane-impermeable drugs.

## 7. Other Natural Hydrogels

### 7.1. Starch-Based Hydrogels (SHs)

Nowadays, a great deal of attention is being paid by biomedical sector researchers to starch-based hydrogels (SHs) [185], as they have some properties similar to those of biological systems [186,187,188,189,190,191]: biocompatibility, high biodegradability and good hydrophobic and mechanical characteristics [192]. Starches are polysaccharides formed by several monosaccharide units linked together by α-D (1–4) and α-D (1–6) bonds [193,194,195,196]. Amylose and amylopectin represent about 98–99% dry weight of starch [197,198]. They have different structures [199,200,201]. Amylose has a linear structure, whereas amylopectin’s structure is branched.

The percentage of amylose and amylopectin in starch is related to the source of the starch (potato, wheat, corn) and to the proportion of the amorphous and crystalline mass [202,203]. The amorphous region in starch mainly consists of amylose. Amylopectin is present in a small amount in the amorphous part, whereas it is the main component of the crystalline region [204]. Even if starches are found abundantly in nature, the usage of native starch presents some drawbacks, such as low dimensional stability, unsatisfactory mechanical properties and low processability of the final products [205,206,207]. Therefore, the use of native starch is not feasible, and the introduction of some modifications by using chemical or/and physical interactions is somewhat desirable [208,209]. To modify SHs, high-energy radiations, such as gamma radiation and electron beam irradiation, can be used with high efficiency and without any contamination [210,211,212,213]. Using high-energy irradiation, homolytic scission of C–H bonds can be obtained. During irradiation, hydroxyl radicals are generated from the water molecules; they remove hydrogen from the starch chain and form starch radicals [214,215]. The swelling power, as well as other properties of the prepared hydrogels, depend on radiation dose and polymer concentration. The described process, due to the mild conditions (room temperature and physiological pH) that can be used, offers many advantages with respect to direct crosslinking. Moreover, the addition of toxic crosslinkers is not necessary.

On the other hand, there have not been many studies carried out in solutions exclusively employing environmentally friendly reagents to produce starch-based hydrogels. Among them, a study by Chin et al. [216] must be mentioned. Starch citrate was produced to exploit natural antibacterial properties. The tricarboxylic acid functional groups of citric acid have the ability to cross the cell membranes of bacteria by suppressing bacterial growth and lowering the intracellular pH of the microorganism [217]. The starch-citrate hydrogel showed great antimicrobial power against *Klebsiella pneumoniae,* demonstrating that it can serve as an alternative antimicrobial agent to fight drug-resistant infection.

### 7.2. Lignin-Based Hydrogels

Lignin (LIG) is a very abundant biopolymer and is an inexpensive, eco-friendly, biocompatible and accessible material.

E. Larraneta et al. [218] proposed an easy way to prepare hydrogels by combining LIG with poly (ethylene glycol) and poly (methyl vinyl ether-co malic acid) through an esterification reaction occurring in the solid state that was significantly accelerated (24 h to 1 h) using microwave radiation (MW). The MW process is shorter, more sustainable and requires less energy compared to the thermal process [219], keeping the production costs low.

Hydrogels were characterized and studied to evaluate their drug release and anti-microbic activity and were compared with lignin [220,221]. It was shown that the hydrogel properties, including the LIG content and the swelling capacity, depended on the molecular weight of PEG. The PEG with a molecular weight of 10,000 allowed the production of a hydrogel with good drug loading and delivery performance. *S. aureus* and *P. mirabilis* are two pathogens that cause infections associated with medical devices. They were used to evaluate the antimicrobial properties of this hydrogel. The results showed that LIG-based hydrogels have higher antibacterial activity than hydrogels without LIG.

### 7.3. Inulin-Based Hydrogels

Inulin-based hydrogels with different degrees of crosslinking density, with the ability to perform the controlled release of 5-fluorouracil (5FU) [222], were manufactured under physiological pH conditions (pH 7.4) in phosphate-buffered saline (PBS). In this environment, the hydroxyl group of inulin between C3–C4 was attacked by sodium periodate (NaIO_4_) and used as an oxidizing agent, causing the breaking of the C-C bond and producing two aldehyde groups. The oxidized inulin crosslinked with non-toxic adipic acid hydrazide AAD and formed a lattice without the use of a catalyst.

### 7.4. Linseed Hydrogels

*Linum usitatissimum* (LUS) mucilage is obtained from the hull of flax seeds by soaking the seeds in distilled water. The main constituent polysaccharides of mucilage are arabinoxylans and rhamnogalacturonans. They exhibit interesting dynamic properties such as swelling (pH-dependent), ability to swell and de-swell according to stimuli, good adhesion, silver reduction and absorption capacity. Mucilage has the ability to swell, and for this reason its hydrogelation potential has begun to be investigated [223]. Due to the low mechanical strength of LUM, a graft copolymerization technique was applied to produce harder, denser materials with higher mechanical strength. Therefore, *Linum usitatissimum* (linseed) mucilage was used as a polymer with acrylamide monomer and methylene bis-acrylamide as crosslinkers for the gel structure.

The properties of the composition were evaluated. Specifically considered were swelling, solvent penetration, and the release of nicorandil, a drug with a very short half-life, that can be considered a strong candidate for release on extended time.

Linseed has been exploited to produce a hydrogel (LUS) for the extended release of antibiotics in the upper digestive tract [224]. LUS does not have high swelling power at pH 1.2, so it was combined with a swelling agent (HPMC) and a channelling agent (β-cyclodextrin) to design a gastroretentive delivery system of antibiotics. To overcome problems associated with drugs soluble in the acidic environment, moxifloxacin was used as a model drug for the realization of a controlled/sustained gastroretentive delivery system, with the main goal of releasing the active compound in a single solution.

The formulations were evaluated in relation to the swelling and flotation properties and to the influence that different concentrations of excipients can have on drug release.

LUS has been successfully used as a reducing and capping agent in Ag-NPs synthesis, to avoid complications, such as Ag-NP coagulation, that usually arise. The phytochemical results of the study performed on an aqueous extract of *Linum usitatissimum* (LUS) by Imran et al. [225] suggest that it can be used for the generation of eco-friendly NPs with antioxidant and other biological applications. Ag-NPs were prepared after being optimized through response surface methodology (RSM), in terms of parameters such as concentration of AgNO_3_, LUS suspension percentage, sonication time and exposure to sunlight. The NPs were tested and confirmed to act as an antimicrobial agent, and also against *E*. *coli* and *B*. *subtilis*, exhibiting anti-biofilm activity. Moreover, its antioxidant power was evaluated by DPPH assay. These biological activities suggest that, by applying small amendments, this system is a suitable candidate for various biological applications.

### 7.5. Gellan-Gum-Based Hydrogels

Gellan gum (GG) is a linear negatively charged exopolysaccharide at physiological conditions. The repeating unit contains four different residues: two D-glucose c, one L-rhamnose and one D-glucuronic acid [226]. GG presents satisfactory biocompatibility, hydrophilicity gelling properties and ability to establish interactions with biomolecules carrying positive charges [227]. Its hydrogels can be exploited for many biomedical applications. Pacelli et al. [228] manufactured a GG/laponite biocompatible composite hydrogel, which was loaded with ofloxacin, acted as a wound dressing and showed mechanical stability upon steam sterilization.

Srisuk et al. [229] combined GC and polyaniline. The obtained hydrogel was shown to be stable, nontoxic, electroconductive and exploitable in regenerative medicine applications. Berti et al. [230] performed polymerization of pyrrole in situ by chemical oxidation, producing electroactive GG/PPy spongy-like hydrogels that can be employed in muscle tissue engineering.

Electro-stimulated drug release devices (EDRDs) are engineered systems that, in response to an electric signal, can release loaded drugs. These devices are usually placed on the skin [231] or implanted in subcutaneous tissue. They are suitable to perform the on-demand administration of drug doses, fulfilling the requirements of treatment protocols. Salas et al. [231] used green routes to prepare electroactive gellan gum (GG)/polypirrole (PPy) nanocomposite hydrogels. PPy colloids were incapsulated within bioamine-crosslinked GG networks. The Ibuprofene (IBP) release kinetic from GG/PPy hydrogels was activated by a pulse potential of 5 V, increasing the drug delivery from hydrogel up to 63%, in contrast to the low IBP amount released in absence of electrical stimuli (10%). The switching behaviour of GG/PPy composite hydrogels with electric pulses makes these materials appealing for electro-stimulated drug-delivery applications.

## 8. Genipin, a Natural Crosslinker

Genipin is an aglycone derived from an iridoid glycoside, called geniposide, found in the fruit of *Gardenia jasmindides* Ellis. Geniposide is an excellent natural crosslinking agent for crosslinking proteins, collagen, gelatine and chitosan. It has much less toxicity than glutaraldehyde and many other synthetic crosslinking regents that are commonly used. Erdagi et al. [232] formulated biocompatible hydrogels by crosslinking a modified diosgenin (DGN) with Genipin. DGN, derived from natural plants, was modified through a conjugation on backbone carboxylated modified nanocellulose to form DGN-NC. Genipin crosslinked gelatine incorporating DGN-NC hydrogels showed good swelling behaviour, depending on DGN-NC concentrations. The results of mechanical property analysis reported a good mechanical integrity for the hydrogels. Neomycin exhibits efficient antimicrobial power when it is loaded onto hydrogels. These hydrogels, thanks to the inhibition of microbial activity which they offer, have been hypothesized to be useful for anti-infective wound protection and for the reduction of healing process time. A two-manner release profile emerged from studies: a rapid release within the initial 15 min and then sustained release for 9–12 h. All the experimental evidence suggests that the crosslinked Genipin/DGN-NC-based hydrogels can act as an antibiotic substrate in wound-dressing applications.

Genipin has also been exploited in the preparation of crosslinked hydrogels devoted to the sustained release of drug in the eye cavity [233,234], improving precorneal residence duration without any eye irritation. Brinzolamide (BZ) is a carbonic anhydrase inhibitor-II, which controls the intraocular pressure (IOP) by decreasing aqueous humour. A hydrogel (BZ-NLC-HG) for sustained ocular delivery has been developed [235]. The procedure occurred in various steps. NLC is the nanolipid-structured system for loading BZ. Before being loaded and incorporated in a hydrogel matrix (HG), the BZ-NLC was prepared and characterized. The dual-sensitive hydrogel HG was realized by a green methodology in which Genipin was used as a crosslinker agent and reacted with carboxymethyl chitosan (CMC)/poloxamer 407 to form a hydrogel. The swelling ratio of (BZ-NLC-HG) hydrogel was analysed in different pH and temperature conditions. The final hydrogel was compared with commercially available eye drops. The results showed that the drug released for a longer duration (24 h) than the commercial product (8 h), and an important benefit was transcorneal permeability 4.54 times greater than that of the marketed eye drops. Moreover, the new hydrogel exhibited no ocular irritation, making it suitable for drug delivery in the treatment of glaucoma.

## 9. In Situ Forming Injectable Hydrogels

Injectable hydrogels, due to properties such as tissue adhesion and sensitivity to pH, are an important challenge for the treatment of diseases requiring a localized drug release [236,237]. The mechanical properties of injectable hydrogels have to be adjusted for their function and applications, to withstand the deformations occurring in the body [238]. They depend on structural parameters, such as porosity (e.g., the space between crosslinks). To promote the performance and integrity of the hydrogel, it is necessary to improve its mechanical strength; this can be realised by increasing the crosslinking degree [239,240]. The aqueous solutions containing the reagents should possess sufficiently low viscosity to allow an easy injection [241,242,243]. This requirement is fulfilled by controlling molecular weight, chemical composition and polymer architecture, properties that are important to ensure even dispersion of the active principle before the gelation process. Clinical applications of injectable hydrogels require also suitable mechanical and viscoelastic properties. Additionally, a biocompatible injectable hydrogel suited for clinical applications should be biodegradable, biocompatible, stable and non-toxic. Above all, since any accumulation could generate adverse effects, it must be taken into account that hydrogels can gradually transform into biocompatible products. Polysaccharides, peptides and nucleic acids spontaneously decompose in non-toxic by-products [242]. Hereafter, some hydrogels developed for local drug delivery or for tissue engineering are reported. Traceability is a prerequisite to carry out targeted monitoring of how the drug is released in situ and to follow the progress and efficacy of therapy. Although fluorescent dyes have already been used for imaging hydrogels, their cytotoxicity limits their applications. A theranostic system must provide non-invasive real-time traceability. Using sericin, a naturally occurring photoluminescent protein from silk, a fully biodegradable and biocompatible injectable sericin/dextran hydrogel was successfully synthesized [243]. It was shown that the hydrogel loaded with doxorubicin significantly inhibited tumour growth.

Chitosan was chemically modified by Mahanta et al. [119] through grafting of polyurethane (PU) to change its highly hydrophilic nature. Urethane linkages in PUs are similar to the peptide ones; an important characteristic that allows their insertion in the human body. Chitosan can be connected to hydrophobic disocyanate polyurethans through bonds formed by using -NH2 and -OH groups present on its backbone. The chemically modified chitosan is less hydrophilic than pure chitosan, as contact angle measurement demonstrated, and shows poor swelling in an aqueous environment. The hydrogel’s porous structure allows better loading and preservation of the active compounds from the surrounding environment. Guaresti et al. [244] successfully synthesized a water-soluble in situ crosslinked hydrogel formed under physiological conditions by Michael-type addition between thiol-modified chitosan and poly (propylene oxide)–poly (ethylene oxide)–poly (propylene oxide) (PPO–PEO–PPO) based bis-maleimide (BMI). The final properties could be modulated by modifying the thiol/maleimide ratio. An increase in the hydrogel storage modulus, as well as in the structure rigidity, with consequent reduction of the water-regaining capacity, was correlated with a higher amount of crosslinker. Thus, the rheological and swelling behaviour were controlled by altering the amount of crosslinking agent in the networks. Moreover, the structure was found to be sensitive to pH stimulus and to specific enzymes, such as lysozymes, suggesting a good response of the hydrogel under human body conditions. Lu et al. [245] realised a hydrogel with antimicrobial, tissue adhesion and haemostatic properties. Their hydrogel was based on glycol chitosan (GC) conjugated with 3-(4-hydroxyphenyl) propionic acid. The compounds gelled in situ immediately after exposition to blue light in the presence of ruthenium complex. The properties of hydration, tissue adhesion, degradability and antimicrobial action of the phenolic GC hydrogel were investigated. Applying the hydrogel to a wound in a mouse liver model, bleeding was greatly and speedily reduced. The GC hydrogel obtained promising results that suggest its usage as a drug releaser, tissue adhesion promoter and haemostat. Khanmohammadi et al. [246] recently obtained injectable hydrogels through enzymatic crosslinking (HRP enzyme, horseradish peroxidase in H_2_O_2_) of two natural molecule units through carbodiimide mediation. They proposed the introduction of phenolic hydroxyl groups (Ph) into neat chitosan (CSPH) and hyaluronic acid (HYAPH), and then the phenol groups were oxidized by the enzymes and formed free radicals that crosslinked through carbon-to-carbon and carbon-to-oxygen (C-C, C-O) bonds, thus allowing the novel hydrogel to be stable [247]. The hydrogel-forming process was tested for different ratios of the two-phenol modified polysaccharides. Increasing HYAPH content, a reduction of the gelation time, of water contact angle and a reduced degradation of hydrogels were observed, whereas compressive modulus and strength increased. The system was used to incorporate chondrocytes for applications concerning cartilage tissue regeneration. The cellular studies showed that mesenchymal stem cells (MSCs) could undergo robust chondrogenesis and generate the proper cartilage scaffolds. Mechanical tests also suggested that the system in future may have the potentiality to be employed to repair cartilage defects.

## 10. Conclusions and Perspectives

The progress of sustainable materials to be employed for biomedical applications depends on the development of new paths to surpass technological challenges. Sustainable green routes must be preferred in the preparation of nano- and microscale composite materials. At the same time, the reduction of the usage of hazardous compounds has to be promoted. Important characteristics of nanomaterials include surface reactivity, colloidal dispersibility and enhanced stability. These requirements are fulfilled by researchers through the use of biopolymers that, owing to their abundance and accessibility, seem to be viable candidates for several biomedical applications, such as tissue engineering, healing of wounds and drug delivery. This review addresses recent studies on hydrogel-based systems, obtained through various green synthetic and technological approaches, grouping them according to the biopolymeric main component used, and the preparation technology process for the micro- or nanoparticles (Table 1).

Each hydrogel platform has its drawbacks and limitations. The most promising applications for drug delivery come from hydrogels formed in situ by simple mixing therapeutic agents and precursor solution, because they seem to produce minimal pain for patients. The rate of the gelation process must be carefully adjusted to avoid inhomogeneous distribution of the drug inside the hydrogel or the diffusion of the precursor components into tissues different from the target one. The viscosity during the injection process and the degradation period have to be controlled. The porosity and the crosslinking ratio of hydrogels have to be carefully determined. The introduction of sustainable theranostic agents inside the hydrogel systems allows better performances to be realized and to achieve platforms aimed for both diagnostic and therapeutic applications. Even if successful experiments have been performed in vitro, their results have to be confirmed by in vivo experiments. Hurdles can always be present when translating from the experimental level to therapeutic applications. Animal models can test short-term biocompatibility, but these tests cannot assess long-term biocompatibility. In vivo studies should confirm the efficacy of each developed drug delivery system, as well as the diagnostic efficiency of the theranostic agents. The deep knowledge and the control of drug delivery carriers in the future will allow the needs of different patients to be met and to personalize therapies. The tuning of hydrogel properties to achieve these results will also require investigations of the release kinetics and trigger conditions.

Long-term biocompatibility is a big challenge for the exploitation of nanocellulose hydrogels in various, especially biomedical, applications. Studies on LCA (Life-Cycle Assessment) of nanocellulose hydrogels are few, and therefore information on the topic is limited. For this reason, it is difficult to understand whether nanocellulose hydrogels may be a risk for various aspects such as biocompatibility. All the processes of biomass production and separation of cellulose from other compounds in the biomass, such as lignin and hemicelluloses, are included in the nanocellulose hydrogel cycle of life. Additionally, solvent and hydrogel production, consumption and biodegradation are included in the life-cycle of nanocellulose hydrogels.

Each procedure should be included in hydrogel LCA studies to assist the identification of alternative resources and feasible procedures that can result in lower environmental and human impacts.

High cost and energy consumption in nanocellulose hydrogels production represent another important challenge. These factors inhibit the widespread adoption of these technologies in most developing countries. Researchers are exposed to the demands of the market and product value chain, so they will have to consider this issue in the future to adapt processes to an industrial scale, not only for nanocellulose hydrogels but for hydrogels in general.

## Figures and Tables

**Figure 1 molecules-28-02107-f001:**
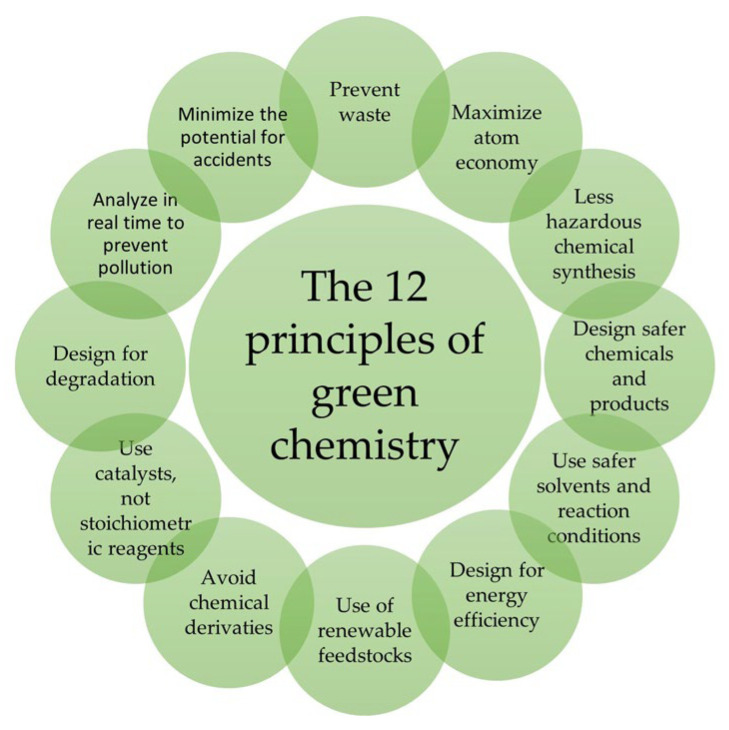
Scheme of the 12 principles of Green Chemistry.

**Figure 2 molecules-28-02107-f002:**
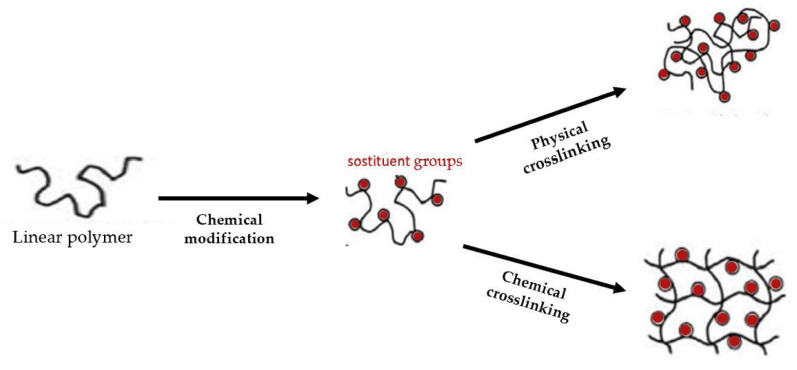
Scheme of chemical and physical hydrogel formation from a linear polymer.

**Figure 3 molecules-28-02107-f003:**
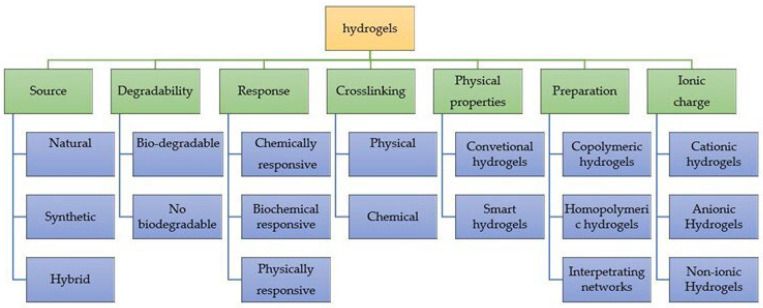
Representative scheme of the classification of hydrogels.

**Figure 4 molecules-28-02107-f004:**
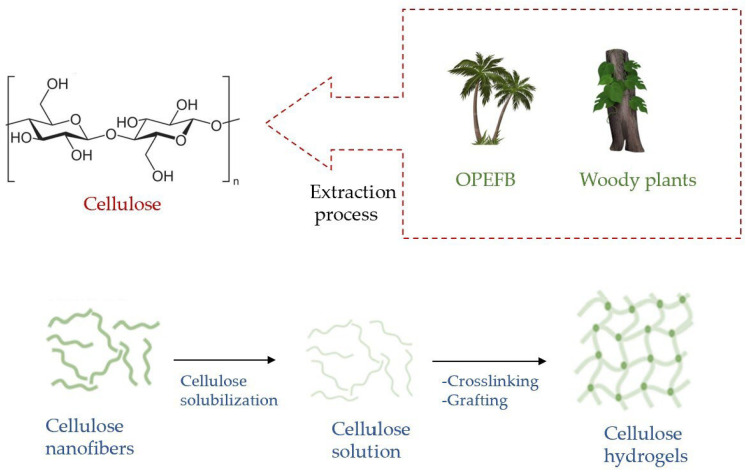
Sources of the extraction of cellulose and the formation of hydrogels.

**Figure 5 molecules-28-02107-f005:**
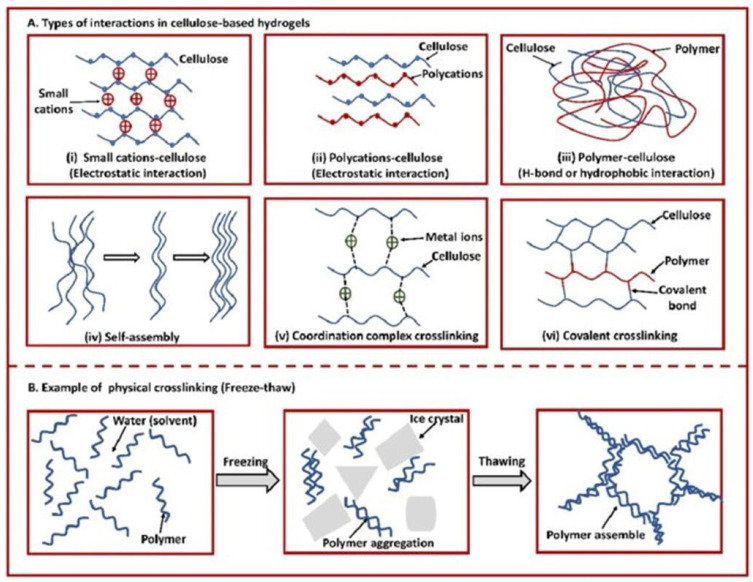
(**A**) Interactions in cellulose-based hydrogels in different systems. (**B**) representation of physical crosslinking, in particular freeze-thawing technique. (**A**,**B**) are reproduced from [58] (license CC BY 4.0).

**Figure 6 molecules-28-02107-f006:**
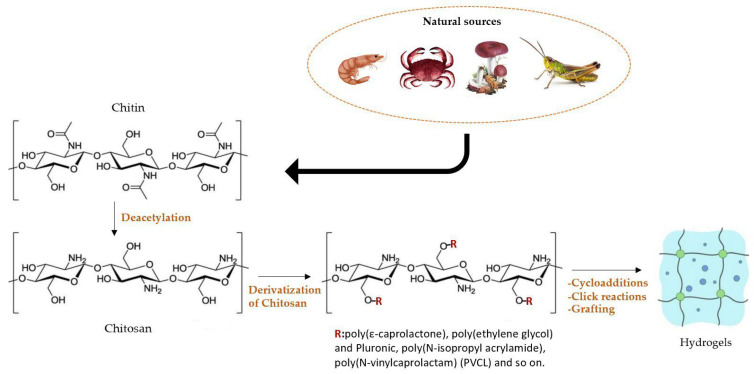
Chitosan molecule, derivatization and hydrogel production.

**Figure 7 molecules-28-02107-f007:**
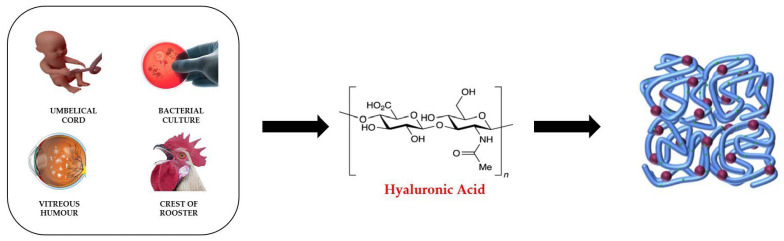
Hyaluronic acid molecule, sources and hydrogel production.

**Table 1 molecules-28-02107-t001:** Grouping of hydrogels based on the biopolymeric main component, and the technological process followed for their preparation. Acronyms used inside the table are described in the text.

Natural Polymer	Technological Process	Refs.
Cellulose	Production of Sustainable Cellulose (OPEFB)	[20,21,22,23]
	Cellulose Solubilization	[52,53]
	Nanocellulose: CNF, CNC, BNC	[35,36,37,38,39,40,41,42,43,44,45,46,47,48]
	Crosslinking: CMC + PVA, hemicellulose adsorption onto CNF, BNC hydrogels	[54,55,56,57,58,59,60,61,62,63,64,65,66,67,68,69,70,71]
	Grafting: Acrylic acid on BNC fibres	[55,75]
	Grafting to CNCs surface	[76]
	CNF + Ag NPs, CNF+ Ca^2+^, Fe_3_O_4_ nanoparticles + β-CD/cellulose	[77,78,79]
Chitosan	Chitin deacetylation	[85,86,87,88]
	Click reactions	[103,104,105,106]
	Diels–Alder Cycloaddition	[105,108,109]
	Huisgen’s reaction	[107,108,109,110]
	Derivatization through IL	[99,100,101,102]
	Grafting: CMCS + MCP, Michael reaction by PEGA, esterdiol polyurethane’ hydrophilic chitosan	[116,117,118,119]
	Beads (coated by cellulose, Silk Fibroin), microbeads through ILs, chitin microbeads	[120,121,122,123,124,125,126,127]
	Radiation Synthesis	[128,129,130,131,132,133]
	Metal–Hydrogel NPs	[73,98,134,135]
Hyaluronic Acid	Modifications for hydrophobicity and bioactivity	[138,139]
	Coupling HYA-Tyramine mediated by Carbodiimide	[141,142,143,144,145]
	Crosslinking in solid phase	[146]
	IEDDA click reaction	[147,148]
	Crosslinking aimed for Tissue Engineering	[149,150]
Alginate	Crosslinking by cations to biopolymers	[151,152,153,154,155,156,157,158,159]
	Metal Nano- and Microparticles (Ag-NPs, HA, MSN)	[160,161,162,163,164,165,166,167,168,169,170,171,172,173,174,175,176,177,178,179,180,181,182,183,184]
Lignin	Lignin + PEG + GAN and MW radiation	[219,220,221,222]
Inulin	Oxidation by periodate, then crosslinking with AAD, accelerated by MW	[223]
Gellan Gum	Electroactive GG crosslinked to amines, such as polyaniline, and incorporating PPy to realize EDRDs	[227,228,229,230,231,232]

## Data Availability

Not applicable.

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
