# Peer review of "Green Chemistry Principles for Nano- and Micro-Sized Hydrogel Synthesis"

_molecules, 2023, doi:10.3390/molecules28052107_

Round 1

Reviewer 1 Report

Dear Authors,

-What are the 12 principles of green chemistry? You should mention about them more detaily or cite more related articles. Also please draw a diagram to highlight these principles.

- "The new frontier for recycling [13,14], “Food Supply Chain Waste” (FSCW), which 78 includes residues generated in the food production/utilization chain, is an exciting chal-lenge to develop new technologies and new methods for the sustainable disposal and re- use of food waste. " this sentences is not good to understand.

-In the introduction part, you should highlight your aim, content or perspective of this review.

-How does the hydrogel look like? You should add a brief figure.

-What does figure 2 tell us? How does cellulose structure involve into the hydrogel?

-You gave only one example to synthesize/ extract cellulose to form a hydrogel. But it is not enough to understand the role of cellulose in hydrogel formation. What does cellulose add to the hydrogel?

-Why did you jump into the solubilization after the production? What is the main issue?

-Then what about the nanocellulose?

-Why did you jump out crosslinking, grafting (synthesis, modification processes) again?

-You dont have any chemical interaction related drawings. It is very hard to understand chemical interactions of integrations and modifications. 

Everything is similar with the chitosan part anf others. The flow of manuscript is not clear. You need to revise all manuscript in order to correct the flow. You dont have any summarizing figure, table or graphical abstract. Also you need to add more updated references. You should discuss more detaily in your conclusion part, add more future directions on the hydrogel research in several fields or focus on basicaly one field. 

Therefore, I suggest that you need the revise whole manuscript.

Regards,

Author Response

Dear Reviewer,

we appreciated your comments and we corrected point by point the manuscript accordingly.

Your comments are in bold text and our responses in plain italics.

Replies to Reviewer 1 comments:

-What are the 12 principles of green chemistry? You should mention about them more detaily or cite more related articles. Also please draw a diagram to highlight these principles.

As reviewer suggested we added a new reference [11b], mentioned about them more detaily (lines 105-109). Moreover we added a diagram, which in the new version of the diagram is indicated as Figure 1, to highlight these principles

- "The new frontier for recycling [13,14], “Food Supply Chain Waste” (FSCW), which 78 includes residues generated in the food production/utilization chain, is an exciting chal-lenge to develop new technologies and new methods for the sustainable disposal and re- use of food waste. " this sentences is not good to understand.

We changed this sentence (lines 113-116) as follows:

The new frontier for recycling [13,14], also known as “Food Supply Chain Waste” (FSCW), is an exciting challenge to develop new technologies and new methods for the sustainable disposal and reuse of food waste. Food waste occurs after food has spoiled or expired due to poor stock management or neglect. It can also occur because of the bad functioning of the food production. This could be due to managerial and technical limitations, such as a lack of proper storage facilities, cold chain and packaging.

-In the introduction part, you should highlight your aim, content or perspective of this review.

We added lines 131-138 to highlight the aim, content and perspectives of our review:

This review aims to highlight hydrogels based on the most abundant and available natural polymers. Attention has been given to preparation processes that do not involve the use of solvents and toxic chemical reagents. In contrast, materials produced by methodologies that contravene the 12 principles of Green Chemistry and are not environmentally friendly, resulting in the waste of important resources such as water, have not been included. It is easily foreseeable that in the near future, research in the field of hydrogels for biomedical applications will increasingly turn toward the exploitation of biopolymers obtained through methodologies that avoid wasting resources and respect the environment.

-How does the hydrogel look like? You should add a brief figure.

We inserted a more detailed figure (Figure 2)  about hydrogel structures a

-What does figure 2 tell us? How does cellulose structure involve into the hydrogel?

We are sorry for the mistake. We have changed the title of the paragraph 3.1 (line 227) and the caption of figure 2 (in the new version of the manuscript it is Figure 4). The figure highlights the plant sources from which cellulose is extracted (see below).

-You gave only one example to synthesize/ extract cellulose to form a hydrogel. But it is not enough to understand the role of cellulose in hydrogel formation. What does cellulose add to the hydrogel?

In this review we wanted to put attention on the most sustainable procedures to extract biopolymers. Paragraph 3.1 treats the extration of cellulose by agricultural biomass and more specifically, the extraction from OPEFB. Nazir et al [36] showed that it is possible produce hydrogels using cellulose extracted from OPEFB.  This method has been illustrated in particular application [34]. In this version of the manuscript we cited the review by Podzil et al., in wich other applications are illustrated and the role of nanocellulose in hydrogels is described (lines 256-260).  In paragraph 3.2., (refs.44,45) some hydrogels based on Nanocellulose (CNF and CNC) extracted from OPEFB are described

In the new version of the manuscript, the paragraph concerning the production of nanocellulose (3.2) precedes the one in which its solubilization (3.3) is discussed. This was done thanks to the referee's suggestions, who invited us to reflect on this point. Solubilization problems affect any kind of cellulose, even nanocellulose, regardless of how it was extracted, in more or less sustainable ways. Sections 3.1, 3.2 and 3.3, have been mostly revised in light of the referee's comments to better clarify the issues of cellulose production. The following sections discuss aspects of hydrogels production and the chemical changes that enable cellulose to become the main component of a hydrogel.

-Why did you jump into the solubilization after the production? What is the main issue?

Thank you for your remark. In the light of the above, we have inverted the position of paragraph 3.2, solubilization of cellulose, with paragraph 3.3, nanocellulose. We have discussed about this isuue in the previous point

-Then what about the nanocellulose?

The nanocellulose paragraph has been moved and supplemented, hopefully the new changes will meet the referee's requests

-Why did you jump out crosslinking, grafting (synthesis, modification processes) again?

In the sections concerning the extraction, preparation, synthesis, and solubilization of cellulose, and more specifically of nanocellulose, we have given only a few mentions  concerning the preparation of hydrogels. In the following sections we go into more specifics about how hydrogels are set up, that is, about the technological processes summarized in Table 1. This has been done for cellulose, but also for the other natural polymers. We regret that our focus has not been identified, and we hope that the updated and extensively revised version of the manuscript will make everything clearer

-You dont have any chemical interaction related drawings. It is very hard to understand chemical interactions of integrations and modifications. 

images have been added that will hopefully provide insight into the chemical interactions that occur in hydrogels.

Everything is similar with the chitosan part anf others. The flow of manuscript is not clear. You need to revise all manuscript in order to correct the flow. You dont have any summarizing figure, table or graphical abstract. Also you need to add more updated references. You should discuss more detaily in your conclusion part, add more future directions on the hydrogel research in several fields or focus on basicaly one field. 

We  answered this issue previously, hopefully, satisfactorily, when we discused about cellulose. As far as we are concerning with chitosane, we carried out the subdivision of the topic into more paragraphes. We hope that also for the rest of the topics, too, restructuring of  the manuscript will make the flow clearer. Some new references have been added. Conclusions have been extensively modified. Great emphasis has been placed on arguing for future perspectives as it was requested.

Reviewer 2 Report

Review of Manuscript ID: molecules-2163274

Journal: Molecules

The review titled: “Green Chemistry Principles for Nano and Micro Sized Hydro- 2 gels Synthesis”, aims to present a brief account of green-manufactured hydrogels and their importance in the field of nanomedicine.     The manuscript needs major revision.   Comments

1)    The language of the manuscript needs major revision, it is very weak. The manuscript contains a lot of typographical errors. Many sentences need rephrasing and should be summarized. The punctuation is missing in many sentences. There is a lot of repetition in many paragraphs.

2)     The manuscript should be summarized in many sections. In addition, the sentences in the manuscript are too long. More shorter sentences must be used in order to prevent the language mistakes.

3)    The abstract should be revised and rewritten, to be more related to the content of the review, and the language must be revised.

4)    I am going to list some of the sentences which require revision and correction. But offcourse the whole manuscript must be revised in details. The review contains a lot of mistakes similar to the below mentioned ones.

Line 21 (revise the sentence), 28-29 (revise the sentence), line 41 (fifteen should be fifty), line 73 (et al.l levels should be at all levels), line 73 (it be toxicity should be it is toxicity), line 114 (check the sentence), line 142 (methods), line144, line 179 (add the reference number), line 205-207 (rephrase), line 220 (full stop missing), line 221 (correct to et al), line 239-242 (there is a repetition in the sentence), line 249-250 (revise), line 252-253 (this sentence was mentioned previously), line 259 (revise the sentence), line 268 (correct: and polymers), line 274 (revise and correct), 276 (repairers????, please correct), line 279 (correct: Polyvinyl alcohol, physically), line 281 (revise the sentence), line 292-294 (revise the language), line 295-298 (the language is weak, rephrase), line 309 (add references), line 314 (revise and correct), line 322 (correct the verb), line 328, line 331, 332, 335, 336 (revise the sentence), line 354 (correct to: performed), 358-359 (revise).

5)    The authors should include several schematic presentations, for each polymer or main title discussed in the review. The authors only included the structure of the natural polymer. In addition, the resolution of all Figures need major improvement.

6)    The manuscript should be revised in order to delete any repetition in the given text.

7)    Title 7.4. Genipin Based Hydrogels is not suitable under the general Title 7 (Other natural hydrogel), as genipin is used as linker and it is not a natural polymer compared to the other polymers discussed in the Section.

8)    The conclusion and perspective need to be rewritten highlighting the future perspectives. This part is very general.

9)    The references needs major revision. The format of the reference is not uniform, one can find different reference format. Example: 136, 137, 142, 144, 147, 148. Please check all references.

Author Response

Dear Reviewer,

we appreciated your comments and we corrected point by point the manuscript accordingly.

Your comments are in bold text and our responses in plain italics.

Replies to Reviewer 2 comments:

-The language of the manuscript needs major revision, it is very weak. The manuscript contains a lot of typographical errors. Many sentences need rephrasing and should be summarized. The punctuation is missing in many sentences. There is a lot of repetition in many paragraphs.

We thank the reviewer for his comments. Taking into account what he indicated, we revised the entire manuscript, revised the English, removed repetitions and introduced punctuation.

-The manuscript should be summarized in many sections. In addition, the sentences in the manuscript are too long. More shorter sentences must be used in order to prevent the language mistakes.

We modified as reviewer suggested

-The abstract should be revised and rewritten, to be more related to the content of the review, and the language must be revised.

Abstract was completely rewritten paying attention to English as was recommended by the referee

-I am going to list some of the sentences which require revision and correction. But offcourse the whole manuscript must be revised in details. The review contains a lot of mistakes similar to the below mentioned ones.

-Line 21 (revise the sentence), 28-29 (revise the sentence), line 41 (fifteen should be fifty), line 73 (et al.l levels should be at all levels), line 73 (it be toxicity should be it is toxicity), line 114 (check the sentence), line 142 (methods), line144, line 179 (add the reference number), line 205-207 (rephrase), line 220 (full stop missing), line 221 (correct to et al), line 239-242 (there is a repetition in the sentence), line 249-250 (revise), line 252-253 (this sentence was mentioned previously), line 259 (revise the sentence), line 268 (correct: and polymers), line 274 (revise and correct), 276 (repairers????, please correct), line 279 (correct: Polyvinyl alcohol, physically), line 281 (revise the sentence), line 292-294 (revise the language), line 295-298 (the language is weak, rephrase), line 309 (add references), line 314 (revise and correct), line 322 (correct the verb), line 328, line 331, 332, 335, 336 (revise the sentence), line 354 (correct to: performed), 358-359 (revise).

We revised everythingh as reviewer suggested

-The authors should include several schematic presentations, for each polymer or main title discussed in the review. The authors only included the structure of the natural polymer. In addition, the resolution of all Figures need major improvement.

the images have been edited and implemented with information.

.

- The manuscript should be revised in order to delete any repetition in the given text.

 We removed all ripetitions.

-Title 7.4. Genipin Based Hydrogels is not suitable under the general Title 7 (Other natural hydrogel), as genipin is used as linker and it is not a natural polymer compared to the other polymers discussed in the Section.

We apologize for the mistake. The paragraph on Genipin has been moved. We created a separate paragraph (paragraph 8), so following sections result renumbered.

-The conclusion and perspective need to be rewritten highlighting the future perspectives. This part is very general.

Conclusions have been extensively modified. Great emphasis has been placed on arguing for future perspectives as it was requested.

-The references needs major revision. The format of the reference is not uniform, one can find different reference format. Example: 136, 137, 142, 144, 147, 148. Please check all references.

As reviewer suggested, we have standardized the references, Some new references have been added and the final order has been changed,

Reviewer 3 Report

The manuscript presents a well-established knowledge of hydrogels.

It is worth enriching it with graphic material, in particular the results of micro-room analyses.

It is also worth summarizing and dividing materials due to their properties and possible applications.

Author Response

Dear Reviewer,

we appreciated your comments and we corrected point by point the manuscript accordingly.

Your comments are in bold text and our responses in plain italics.

Replies to Reviewer 3 comments:

The manuscript presents a well-established knowledge of hydrogels.

It is worth enriching it with graphic material, in particular the results of micro-room analyses.

It is also worth summarizing and dividing materials due to their properties and possible applications.

We have revised the manuscript following the indications of the reviewer

Round 2

Reviewer 1 Report

Dear Authors,

Thanks for your revisions.

Everything is more clear now. But all figures that you added have low resolution. Please add more high resoluted (at least 300 dpi ) figures. 

And I suggest you to add a graphical abstract to summarize your whole manuscript's aim. 

Regards,

Author Response

Dear Reviewer,

we appreciated your comments and we added more high resoluted figures and a graphical abstract. 

Reviewer 2 Report

The caption of all Figures must be revised and corrected. The caption should indicate exactly what is presentedin the Figure. The author should correct the caption of Figures 2, 3, and 8.

In addition, the resolution of all Figures should be improved to be able to read the text. 

Author Response

Dear Reviewer,

we appreciated your comments and corrected the captions of figures 2, 3 and 8 and improved the resolution of all figures.